Bovine tuberculosis breakdown duration in cattle herds: an investigation of herd, host, pathogen and wildlife risk factors

Milne Georgina georgina.milne@afbini.gov.uk 1
Allen Adrian 1
Graham Jordon 1
Lahuerta-Marin Angela 1
McCormick Carl 1 2
Presho Eleanor 1
Reid Neil 3
Skuce Robin 1
Byrne Andrew W. andreww.byrne@agriculture.gov.ie 1 3 4
1 Veterinary Sciences Division, Agri-food and Biosciences Institute , Belfast , United Kingdom
2 Department of Agriculture, Environment, and Rural Affairs , Coleraine , United Kingdom
3 School of Biological Sciences, Queen’s University Belfast , Belfast , United Kingdom
4 One-Health Scientific Support Unit, Department of Agriculture, Food and the Marine , Dublin , Ireland
Alvarez-Rodriguez Javier
Electronic publication date: 2020 Feb 3
Publication date: 2020
Volume: 8
Electronic Location ID: e8319
Received 2019 Sep 11; Accepted 2019 Nov 29
Copyright: ©2020 Milne et al.
Copyright year: 2020
Copyright holder: Milne et al.
License: This is an open access article distributed under the terms of the Creative Commons Attribution License, which permits unrestricted use, distribution, reproduction and adaptation in any medium and for any purpose provided that it is properly attributed. For attribution, the original author(s), title, publication source (PeerJ) and either DOI or URL of the article must be cited.
License URL: https://creativecommons.org/licenses/by/4.0/

Keywords: Bovine TB, Badger, Cows, Northern Ireland, UK, Veterinary epidemiology

Funding: Department of Agriculture, Environment and Rural Affairs (DAERA) 48005 (122035): 15/3/10 This work was supported by Department of Agriculture, Environment and Rural Affairs (DAERA) (Grant: 48005 (122035): 15/3/10 “The evaluation of the role of multiple reactor and chronic breakdown herds in the epidemiology of bovine tuberculosis in Northern Ireland”). The funders had no role in study design, data collection and analysis, decision to publish, or preparation of the manuscript.

==============================
Background

Despite rigorous controls placed on herds which disclose ante-mortem test positive cattle to bovine tuberculosis, caused by the infection of Mycobacterium bovis, many herds in Northern Ireland (NI) experience prolonged breakdowns. These herds represent a considerable administrative and financial burden to the State and farming community.

Methods

A retrospective observational study was conducted to better understand the factors associated with breakdown duration, which was modelled using both negative binomial and ordinal regression approaches.

Results

Six explanatory variables were important predictors of breakdown length in both models; herd size, the number of reactors testing positive in the initial SICCT test, the presence of a lesioned animal at routine slaughter (LRS), the count of M. bovis genotypes during the breakdown (MLVA richness), the local herd-level bTB prevalence, and the presence of herds linked via management factors (associated herds). We report that between 2008 and 2014, mean breakdown duration in NI was 226 days (approx. seven months; median: 188 days). In the same period, however, more than 6% of herds in the region remained under movement restriction for more than 420 days (13 months); almost twice as long as the mean. The MLVA richness variable was a particularly important predictor of breakdown duration. We contend that this variable primarily represents a proxy for beef fattening herds, which can operate by purchasing cattle and selling animals straight to slaughter, despite prolonged trading restrictions. For other herd types, the model supports the hypothesis that prolonged breakdowns are a function of both residual infection within the herd, and infection from the environment (e.g. infected wildlife, contiguous herds and/or a contaminated environment). The impact of badger density on breakdown duration was assessed by including data on main sett (burrow) density. Whilst a positive association was observed in the univariate analysis, confounding with other variables means that the contribution of badgers to prolonged breakdowns was not clear from our study. We do not fully reject the hypothesis that badgers are implicated in prolonging bTB breakdowns via spillback infection, but given our results, we posit that increased disease risk from badgers is unlikely to simply be a function of increasing badger density measured using sett metrics.

Introduction

Bovine tuberculosis (bTB), caused by Mycobacterium bovis bacterial infection, presents an ongoing epidemic in many countries (Humblet, Boschiroli & Saegerman, 2009). In Britain and Ireland, bTB remains stubbornly persistent despite long-term and intensive programs focusing primarily on controlling bTB breakdowns in cattle herds (Allen, Skuce & Byrne, 2018). In Northern Ireland (NI) infection levels remain high, with an annual herd level incidence of over 8% (DAERA, 2018). Current bTB controls are compliant with EU Directive 64/432/EEC (as amended) and consist of a “test-and-slaughter policy”, alongside active routine slaughterhouse surveillance (Abernethy et al., 2006; Abernethy et al., 2013). Herds undergo annual testing using the single intradermal comparative cervical tuberculin (SICCT) test, with infected herds subsequently placed under trading restrictions (a herd breakdown) until two clear herd level tests are obtained, each not less than 60 days apart (DAERA, 2017). The shortest length of time a herd usually remains under trading restrictions is therefore 120 days. All animals which test positive to the SICCT test are culled. To aid in the detection and eradication of bTB, additional testing can be undertaken in problematic herds using the interferon gamma test (Lahuerta-Marin et al., 2015). Despite these efforts, some herds fail to clear infection upon retest and remain persistently infected, resulting in prolonged or recurrent breakdowns (Doyle et al., 2016; Milne et al., 2019a). As herd-keepers are compensated for culled cattle, the remuneration costs of persistent breakdowns contribute disproportionally to the total program costs, which have exceeded £30 million per annum in recent years (NIAO, 2018). Additionally, the trading restrictions and production losses associated with persistent breakdowns also present considerable economic and emotional burdens to the farming community (Robinson, 2017).

Previous studies from Great Britain (GB) and the Republic of Ireland (ROI) have defined these persistently infected (i.e., “chronic”) herds using a number of non-mutually exclusive criteria. These include recurrence of bTB in a herd (Doyle et al., 2016; Gallagher et al., 2013; Karolemeas et al., 2012; Karolemeas et al., 2011; Wolfe et al., 2010), the prolongation of trading restrictions (Doyle et al., 2016; Griffin et al., 1993; Karolemeas et al., 2012; Karolemeas et al., 2010) and outbreak size (Clegg et al., 2018). Here, we focus specifically on bTB persistence as defined by breakdown length and measured via the duration of movement restriction periods. Indeed, extended periods of trading restrictions have been observed in herds across these islands (More et al., 2018). For example, in the ROI in 2012, 16.8% of herds were restricted for over 255 days (seven and a half months) (Houtsma et al., 2018). In England, 5.8% of breakdowns lasted longer than 550 days (18 months), in contrast to the mean breakdown length of 192 days (AHVLA, 2016). In NI, a previous study found that the median breakdown duration was 184 days (approx. six months) (Doyle et al., 2016); however, between 2003 and 2015, 4.3% of bTB breakdowns in NI were classified as prolonged (>550 days) (More et al., 2018).

Earlier work has enabled better understanding of the factors associated with breakdown duration. A previous study from NI between 2005 and 2010 showed that bTB breakdowns lasting longer than 365 days were associated with local area bTB prevalence, the presence of associated herds (i.e., herds linked via geography, family or some other management factor), the number of years previously restricted, the number of cattle reactors at the disclosing test, the total number of reactors over the outbreak, and the identification of a lesion consistent with bTB at routine slaughter (Doyle et al., 2016). In Great Britain (GB) prolonged breakdowns were particularly associated with the confirmation status of the breakdown and herd size (Karolemeas et al., 2010). Comparisons between transient bTB breakdowns (≤6 months) and breakdowns lasting >6 months (i.e., “persistent”) in GB found that herd size, herd management, and the presence of active badger setts were important explanatory variables associated with bTB persistence (Reilly & Courtenay, 2007). Whilst breakdown prolongation is either a feature of failure to clear infection from the herd (i.e., within herd recrudescence), and/or re-infection from local sources (e.g., contagious herds or a local wildlife reservoir), it is not yet possible to disentangle these various routes of infection.

The European badger (Meles meles) is a well-documented infection reservoir for M. bovis (Byrne et al., 2014b; Gallagher & Clifton-Hadley, 2000) and a number of other studies have also explored the association between bTB breakdown duration and wildlife. In the ROI, the presence of badgers was associated with bTB breakdowns lasting greater than one year (Griffin et al., 1993), and reactive badger culling was related to the prolongation of bTB breakdowns in GB (Karolemeas et al., 2012). However, the contribution that badgers make towards protracted bTB breakdowns is not well understood to date. Furthermore, previous work largely consists of case-control studies, and does not model breakdown length explicitly. The aim of this work, therefore, was to model the factors associated with breakdown duration, including variables associated with M. bovis molecular genotype data (Skuce et al., 2010), badger density data (Reid et al., 2012), alongside herd characteristics. For the first time, breakdown length was modelled as both a continuous and ordinal variable, which we believe improves our understanding of within-herd bTB dynamics and could be applied to bTB management in many endemic regions globally.

Materials & Methods

Study area

Northern Ireland is approximately 14,000 km2. The official bTB control programme is administered over ten Divisional Veterinary Office DVO areas, comprised of 123 “patches”; mean size 110 km2 (SD ± 53); Fig. 1.

Figure 1 The distribution of continuous variables across each DVO area within Northern Ireland.

(A) Breakdown Length; (B) Herd size; (C) Total Reactors; (D) Yearly Patch Prevalence; (E) Total Patch Prevalence; (F) Main Sett Density; (G) Out-moves Before Breakdown; (H) In-moves Before Breakdown. A number of variables exhibited little variation in the median values per-DVO and are therefore not displayed (outbreak reactors, median = 1 for all DVOs; MLVA Richness, median = 1 for all DVOs).

Study design

Two retrospective analyses were undertaken, firstly (i) quantifying the risk factors associated with bTB breakdown duration, using negative binomial (count) regression with the outcome measured in days, and (ii) quantifying the risk associated with bTB breakdown duration using ordinal regression, with the outcome modelled as a categorical ordered variable. This approach was considered necessary to account for bTB breakdown administration in NI. Herd breakdown duration measures arise as a result of a disease management process, and are not a wholly natural phenomenon. Generally, once bTB has been confirmed, the herd Officially Tuberculosis free status is Withdrawn (OTW). Usually, two clear herd tests are required to restore Officially Tuberculosis Free (OTF) status. Each herd-level test is scheduled to occur a minimum of 60 days apart (DAERA, 2018). For the ordinal regression therefore, breakdown duration was classified into four distinct categories based on multiples of 60 days (it should be noted, however, that breakdown length may not always correlate exactly with the number of tests done, as herds may delay testing). The first category contained breakdowns ≤180 days (approx. 6 months; 3 tests until OTF status restored), the second category included breakdowns which ended up to 120 days later; ≤300 days (approx. 9 months; 5 tests until OTF status restored), and the third category included breakdowns which ended up to 120 days after this; ≤420 days (approx. 13 months; 7 tests until OTF status restored). The final category included breakdowns which lasted longer than 421 days (8 or more tests until OTF status restored). Breakdown start dates are denoted by the date at which the first SICCT reactor or lesioned animal identified at slaughter was disclosed, and the breakdown end date was the test at which the last clear herd test was achieved.

Dataset creation

BTB breakdown data spanning January 2003 to December 2015 inclusive (n breakdowns = 27,718) were made available from the NI Department of Agriculture, Environment and Rural Affairs (DAERA) database, the Animal and Public Health Information System (APHIS) (Houston, 2001). This dataset was restricted to only include OTW breakdowns (n = 19,084; 8,634 breakdowns removed), which were defined by policy guidance at the time of study as the presence of more than five SICCT reactors, or two positive results to the four possible bTB tests; confirmation via histopathology, culture or spoligotyping, or the identification of a lesion at routine slaughter. Breakdowns with incomplete or erroneous information were also excluded (e.g., missing GIS information, MLVA information, or breakdowns lacking end dates (n = 17,114; 1,970 breakdowns removed). The dataset was further restricted to include breakdowns which started and ended between 01∕01∕2009 and 31∕12∕2014 (n = 7,478; 9,636 breakdowns removed). These dates were chosen because surveillance using M. bovis MLVA genotyping data occurred at the herd level between 2003 and 2008, but from 2009 onwards, all culture confirmed animal-level M. bovis isolates were genotyped. Finally, breakdowns which were recorded as lasting less than 60 days were excluded from the final dataset (n = 5 breakdowns removed), as 60 days is the minimum restriction period which may be permitted under some circumstances e.g., less than five positive SICCT animals with no post-mortem or laboratory confirmation (DAERA, 2019). The final dataset contained information on 7,473 breakdowns. All data were assembled and analysed using Microsoft Access 2007 (12.0.6735.5000) SP3 MSO and R Version 3.2.5 (R Core Team, 2013).

The fixed-effect variables considered in the analysis are shown in Table 1. They were derived and defined as follows; herd size (number of animals in the herd at the time of breakdown); outbreak reactors (the number of SICCT reactors present in the disclosing test); total reactors (the total number of SICCT reactors during a breakdown); yearly patch prevalence (herd level bTB prevalence for the year); mean patch prevalence (mean herd level bTB prevalence), outward moves year before (the number of outward cattle moves in the year prior to breakdown), and inward moves year before (the number of inward cattle moves in the year prior to breakdown). A categorical herd type variable was included (beef, dairy, other, or unknown). Binary variables were the presence or absence of a milk license, whether lesions consistent with tuberculosis were identified during routine slaughter (LRS), the presence or absence of associated herds (herds are “associated” via e.g., shared management, shared grazing, or shared family responsibilities), and whether the herd had any previous breakdowns during the study period. The herd DVO, the year of breakdown, and the herd unique identifier were included as random effect variables. The distribution of explanatory variables across each DVO in NI is illustrated in Fig. 1.

Table 1 Summary statistics of the fixed effect explanatory variables.

Variable	Count	Median	Mean	IQR (1st-3rd)	
breakdown_length		188	225.6	140–260	
herd_size		93	141.2	42–190	
outbreak_reactors		1	2.84	1–2	
total_reactors		3	7.66	2–8	
year_patch_prev		8.85	9.89	6.01–12.55	
mean_patch_prev		10.18	10.73	7.83–13.24	
MLVA_richness		1	1.25	1–1	
main_sett		0.74	0.77	0.56–0.92	
outwards_moves_year_before	52	98.22	23–106	
inwards_moves_year_before	9	59.78	1–42	
LRS	2,209				
milk_licence	2,360				
associated_herds	1,501				
previous_breakdown	2,061				
herd_type beef	3,617				
herd_type dairy	2,275				
herd_type other	98				

M. bovis MLVA genotype data

M. bovis MLVA genotype data were derived from isolates obtained from skin-test reactors, and from lesioned animals identified at routine slaughter. These animal-level data were then associated with bTB breakdown-level data. From this, breakdown-level metrics of MLVA genotype richness (number of different MLVA types) were calculated. The process of genotyping M. bovis isolates has been described more fully elsewhere (Kamerbeek et al., 1997; Skuce et al., 2010; Skuce et al., 2005). Briefly, all culture-confirmed bTB cases were sub-cultured to single colonies and heat-killed to create PCR-ready bacterial cell lysates. These were then used as PCR templates for molecular characterisation of pathogen variation. Eight VNTR loci across the M. bovis genome were genotyped; MV2163B/QUB11B, MV4052/QUB26A, MV2461/ETRB, MV1955/Mtub21, MV1895/QUB1895, MV2165/ETRA, MV2163/QUB11A and MV3232/QUB3232 (Durr, Hewinson & Clifton-Hadley, 2000).

Badger density

Badger main sett density was incorporated into models by using a data from the Northern Ireland Badger Survey 2007-08 (Reid et al., 2012). This enumerated and mapped badger main setts within 212 regularly spaced 1 km2 squares throughout Northern Ireland, and subsequently spatially interpolated using the Kriging function of the ArcMap 10.5 (ESRI, California, USA) Spatial Analyst toolbox, providing a heat-map proxy of badger density throughout the region (Reid et al., 2012).

Data modelling

During the univariable stage of model fitting for both the count and ordinal models, predictor variables were explored using summary statistics and cross–tabulations with the outcome variable. The relationship between each predictor and the outcome was also visually scrutinised using ggplot2 (Wickham, 2009). Predictor variables were then considered individually for association with the outcome. Correlation coefficients between variables were determined. Variables with moderate or strong correlation ≥ 0.5 or ≤−0.5 were identified, and from these, only those variables with the strongest association with the outcome were retained, based on log-likelihood values. Following univariable assessment, generalised linear mixed models (GLMMs) were fitted. The count model was constructed using the package lme4 (Bates et al., 2015), and the ordinal model was constructed using the package ordinal (Christensen, 2019). Initial modelling of the count data using Poisson regression indicated the presence of over-dispersion (the variance was greater than the mean); a negative binomial model was instead found to be more suitable for these data (Zuur, Hilbe & Ieno, 2015; Zuur et al., 2009).

In both count and ordinal models, the DVO, breakdown year, and herd identifier were included as nested random effects (Zuur et al., 2009). Continuous variables were log-transformed in the final models for computational efficiency, and to improve the model fit (i.e., ensure all explanatory variables were on the same scale, to approximate a more linear relationship, reducing skew and to limit the influence of outliers). All predictors were initially included in the model, including biologically plausible two-way interactions. Final models were assembled using backwards stepwise selection routines; better fitting models were selected on the basis of likelihood ratio tests (Christensen, 2019; Zuur et al., 2009). At each stage, however, model coefficients were manually assessed for confounding (Dohoo, Martin & Stryhn, 2009). Once final models were constructed, excluded predictor variables were again offered to the model and the impact assessed using likelihood ratio tests. Final models were screened for correlations between fixed effects and random effects and were assessed by visual examination of residuals. Plots of residual versus fitted values were firstly explored; residuals were then plotted against all covariates included in the model, and also against the covariates which had been excluded during model fitting. Residuals were used to identify influential data-points, and models were re-run with these data removed for comparative purposes.

Ordinal regression assumes that the effects of explanatory variables are consistent across all outcome categories (i.e., the assumption of proportional odds). We firstly attempted to test this using the nominal_test function of the Ordinal package (Christensen, 2019). However, at the time of analysis, this function was not available for models with multiple random effects. Furthermore, it is presently not feasible to construct an ordered regression model with multiple random effects for which the assumption of proportional odds is also relaxed (Christensen, 2019). To overcome this, we constructed an initial ordered regression model including only fixed effects (via the clm function) and tested the assumption of proportional odds on this model (the nominal_test function). Explanatory variables which violated the assumption of proportional odds were identified and the model was re-ran, wherein the proportional odds assumption was relaxed for these variables. However, the final clmm model was further validated by comparing the model coefficients against those derived from three binary logistic GLMMs (Armstrong & Sloan, 1989; Ananth & Kleinbaum, 1997), with the binary outcome variable dichotomised at the same levels as in the ordinal regression. In these three models, the outcome (breakdown length) was dichotomised as follows: Model 1; ≤180 days (breakdowns 180 days or less classified as 0, all others classified as 1); Model 2 ≤300 days; (breakdowns 300 days or less classified as 0, all others classified as 1); and Model 3 ≤420 days (breakdowns 420 days or less classified as 0, all others classified as 1).

Figure 2 Distribution of breakdown length.

The (A) frequency distribution and (B) cumulative distribution of the breakdown length variable. The three different cut-offs (180 days, 300 days and 420 days) are shown as vertical lines.

Results

Summary data

The final dataset contained 7,473 breakdowns associated with 5,378 herds. The mean breakdown length was 226 days (SD ± 140 days; approx. seven months) and median breakdown length was 188 days (Inter Quartile Range (IQR): 140–260 days; approx. six months). The longest breakdown was recorded at 2,288 days (6 years). When classified into categories, almost half of all breakdowns (47.18%, n = 3,526) lasted less than 180 days. 34.86% (n = 2,605) of breakdowns were between 181 and 300 days in duration, 11.33% (n = 847) lasted between 301 and 420 days, whilst 6.62% of all bTB breakdowns (n = 495) lasted 421 days or longer (13 months; i.e., 8 or more tests were required to restore OTF status). The distribution of the breakdown length outcome variable is shown in Figs. 2A–2B. Mean breakdown duration varied across NI, from a minimum of 192 days in Derry/Londonderry DVO to a maximum of 266 days in Newry DVO (Fig. 1).

Count model results

The results of the count model of breakdown duration is shown in Table 2 (Table S1). The final model contained seven explanatory variables. The exponentiated results are reported here as Incidence Rate Ratios (IRR) with associated 95% upper and lower confidence intervals (CI). The variables log herd size (IRR: 1.05, 95%CI [1.04–1.06]), log outbreak reactors (IRR: 1.05, 95%CI [1.04–1.06]), log mean patch prevalence (IRR: 1.04, 95%CI [1.01–1.07]) and log MLVA richness (IRR: 1.62, 95%CI [1.58–1.67]) were positively associated with breakdown duration. The binary variables for presence of an LRS (IRR: 1.12, 95%CI [1.09–1.14]), presence of associated herds (IRR: 1.10, 95%CI [1.07–1.13]) and a previous breakdown (IRR: 1.04, 95%CI [1.02–1.07]) were positively associated with breakdown duration. Re-running the model with influential data removed resulted in only minimal change in parameter estimates when compared to the original model (<15% change). The addition of a quadratic term for log MLVA richness was also found to significantly lower log-likelihood; this model is shown in Table S2.

Table 2 Parameter estimates of the fixed effectexplanatory variables in the final model for both the count model (negative binomial) and ordinal model.

Variable	IRR	95% CI Lower	95% CI Upper	OR	95% CI Lower	95% CI Upper	
log(herd_size)	1.05	1.04	1.06	1.26	1.20	1.32	
log(outbreak_reactors)	1.05	1.04	1.06	1.34	1.26	1.43	
log(mean_patch_prev)	1.04	1.01	1.07	1.20	1.04	1.37	
log(MLVA_Richness)	1.62	1.58	1.67	7.06	6.04	8.24	
LRS_binary1	1.12	1.09	1.14	1.79	1.59	2.01	
associated_herds_binary1	1.10	1.07	1.13	1.49	1.32	1.69	
previous_breakdown	1.04	1.02	1.07	–	–	–	

Ordinal model results

Six variables were identified as important predictors in the ordinal model. The parameter estimates of the final model are shown in Table 2 (Table S3). All six variables in the final model were found to be positively associated with the increasing breakdown duration; log herd size (OR: 1.26, 95%CI [1.20–1.32]), log outbreak reactors (OR: 1.34, 95%CI [1.26–1.43]), log mean patch prevalence (OR: 1.20, 95%CI [1.04–1.37]), log MLVA richness (OR: 7.06, 95%CI [6.04–8.24]), the presence of an LRS (OR: 1.79, 95%CI [1.59–2.01]) and the presence of associated herds (OR: 1.49, 95%CI [1.32–1.69]). The coefficients derived from this model were similar to a fixed-effect ordinal regression model, however the variables log herd size, log MLVA richness and log outbreak reactors violated the proportional odds assumption (p < 0.05), suggesting that the effect size is not the same across all three breakdown duration categories. As the assumption of proportional odds was not met for all variables, the coefficients from ordinal model were also compared to those derived from three binomial logistic GLMMs (Fig. 3). There was only limited evidence of the parameter estimates differing between ordinal and binomial models. The binomial model of breakdowns lasting 420 days or less returned a higher odds ratio associated with herd size (OR: 1.49, 95%CI [1.32–1.67]) than the ordinal model (OR: 1.26, 95%CI [1.20–1.32]). The parameter estimate for the number of outbreak reactors was elevated in the binomial model of breakdowns lasting less than 180 days (OR: 1.56, 95%CI [1.45–1.49]) compared to the ordinal model (OR: 1.34, 95%CI [1.26–1.43]), and was also diminished in the model of breakdowns lasting 301 days or more (OR: 1.23, 95%CI [1.03–1.23]) and 421 days or more (OR: 1.08, 95%CI [0.94–1.24]).

Figure 3 Comparison of parameter estimates across models.

Comparison of parameter estimates for the six explanatory variables obtained from the ordinal regression model with four categories (full model), compared to parameter estimates obtained from three binary logistic regression models (model type).

Figure 4 Relationship between (A) MLVA genotype richness and categorical breakdown duration; (B) how the number of reactors over a breakdown differs between production types; (C) how the breakdown length differs between production types and; (D) the confounding between the number of reactors over a breakdown and MLVA type, and how these differ between production types.

Figure 5 Correlations between (A) mean breakdown length per DVO and main sett density and (B) mean patch-level bTB prevalence and main sett density. (C) shows the relationship between main sett and breakdown duration on a per-DVO basis, including only data available for each DVO, and (D) is the same as with (C), but without confidence intervals and predicted against the full range of values for all DVOs.

MLVA Genotype richness

MLVA genotype richness was the most important variable in both count and ordinal models, in terms of both effect size and decrease in model deviance. This was particularly observable in the ordinal regression model (Fig. 4A). The MLVA genotype richness variable was moderately correlated with the number of inwards moves in the year prior to breakdown (r = 0.33), outwards moves in the year prior to breakdown (r = 0.34) and the number of total reactors over the breakdown (r = 0.39). Further investigation into this “total reactors” variable revealed significantly more reactors in herds with a milk license (mean = 11) than herds without a milk license (mean = 6; Univariable Negative Binomial Regression, IRR: 1.70; 95%CI [1.62–1.79]; Fig. 4B). However, the presence of a milk license was only ‘marginally significant’ in a univariable analysis of breakdown length in both count (IRR: 1.02, 95%CI [1.00–1.05]) and ordinal models (OR: 1.16, 95%CI [1.01–1.27]), and was not retained as a predictor of breakdown length in the finals GLMMs after model building. Further analysis showed that whilst mean breakdown length in herds with a milk license was indeed marginally longer (230 days  ± 141) than in herds without a milk license (224 days ± 140), some of the longest breakdowns were found in herds without milk licenses. For example, there were 27 breakdowns lasting over 1000 days; 10 were in herds with milk licenses, and 17 were in herds without; Fig. 4C. It would therefore appear that whilst production type per-se is not a useful predictor of breakdown length, the results show that some variables which vary between production types—the number of reactors over a breakdown for example (here, confounded with MLVA genotype richness, Fig. 4D), are indeed important predictors of breakdown length.

Badger density results

When modelled using a univariate negative binomial GLM, badger main sett density was a significant predictor of breakdown length (IRR: 1.13, 95%CI [1.13–1.14]). However, this variable was not retained in the final GLMM. Further investigation found that main sett density was correlated with other explanatory variables. Thus, main sett density per-DVO was moderately correlated with breakdown length per-DVO (r = 0.57) and with breakdown length per-patch (r = 0.32), suggesting that the spatial variables already included in the model, notably DVO, captured the general positive relationship observed between main sett density and breakdown length; Fig. 5A. Furthermore, when compared to a fixed effects univariate GLMs where DVO was the sole predictor of breakdown length, the addition of the main sett density variable did not result in a better fitting model (χ2 = 0.02, df = 1, p = 0.90). An interaction between DVO and sett density was, however, significant when compared to the fixed-effects model with non-interacting DVO and main sett variables (χ2 = 24.24, df = 9, p = 0.004; Tables S4 and S5), suggesting a differential relationship between sett density and breakdown length on a per-DVO basis which was not immediately observable when data were not stratified by DVO. Figures 5C–5D illustrates this observation. Whilst a positive association was found between main sett density and breakdown length in Ballymena, Coleraine, Dungannon, Larne and Derry/Londonderry DVOs, a negative relationship between sett density and breakdown length was observed in Armagh, Enniskillen, Newry, Newtownards and Omagh DVOs (Table S6). To explore this further, we therefore present a second GLMM, (Table 3) in which main sett density was permitted to differ on a per-DVO basis (i.e., a random slopes and random intercepts model). It should be noted, however, that the inclusion of the random slopes term for main sett resulted in only marginally improvements, compared to the original GLMM (Table 2) which did not include a random slope for main sett density per-DVO (χ2 = 4.61, df = 2, p = 0.099)

Table 3 Parameter estimates of the explanatory variablesin an alternative count model, allowing main sett to vary on a per DVO basis.

DVO	main_sett slope	Intercept	
Armagh	−0.030	5.031	
Ballymena	0.039	4.874	
Coleraine	−0.004	4.971	
Dungannon	−0.031	5.033	
Enniskillen	0.047	4.856	
Larne	0.017	4.923	
Londonderry	0.031	4.892	
Newry	−0.107	5.204	
Newtownards	−0.010	4.986	
Omagh	0.022	4.913	
Variable	Est	Std. Error	z	IRR	95%CI Lower	95%CI Upper	
Intercept	4.96	0.04	120.1	143	131.89	155.04	
log(herd_size)	0.05	0.00	9.80	1.05	1.04	1.06	
log(outbreak_reactors)	0.05	0.01	7.57	1.05	1.04	1.06	
log(main_sett)	0.01	0.02	0.39	1.01	0.97	1.05	
log(MLVA_Richness)	0.48	0.01	34.51	1.62	1.58	1.67	
LRS_binary1	0.11	0.01	9.68	1.12	1.09	1.14	
associated_herds_binary1	0.09	0.01	7.21	1.10	1.07	1.12	
previous_breakdown	0.04	0.01	3.57	1.04	1.02	1.07	

Further analysis also indicated that that the main sett density variable exhibited moderate correlation with mean patch prevalence (r = 0.40; Fig. 3C). To better understand the effect of main sett density on breakdown duration in the absence of spatial confounders, two further alternative models were constructed, both omitting DVO from the random effects component and including log main sett in the fixed effects component. These models also incorporated the other fixed-effect variables reported in Table 2, however, one of these models included patch prevalence in the fixed component, and the other did not. In the model which omitted both DVO and log patch prevalence, log main sett was a significant predictor of breakdown length (OR: 1.08, 95%CI [1.05–1.11]; Table S7). Log main sett was also found to be an important predictor of breakdown length when log patch prevalence was included (OR: 1.08, 95%CI [1.05–1.11]; Table S8), however in this model, log patch prevalence was no longer an important predictor of breakdown duration (OR: 1.01, 95%CI [0.99–1.11]; χ2 = 1.02, df = 1, p = 0.31). Confounding between main sett density and DVO was also observed in the ordinal regression. Thus, main sett density was positively associated with increasing breakdown duration categories in a univariable GLM (OR: 1.59, 95%CI [1.42–1.78]), but the main sett variable was not recovered as an important predictor of breakdown length in the mixed model context. We constructed a univariable ordinal GLM with DVO as the sole predictor of breakdown duration category. The coefficients from this model (i.e., the “risk” associated with each DVO) was positively associated with mean sett density per DVO (r = 0.59). Additionally, the inclusion of the main sett variable in this model did not improve model fit (χ2 = 0.04, df = 1, p = 0.84).

Discussion

The heterogeneity in transmission of infections across populations is a well-known phenomenon in many systems (Woolhouse et al., 1997), where a small proportion of the population can contribute disproportionately to disease maintenance. Our work highlights this issue in the context of prolonged bTB breakdowns. The results show that mean breakdown length was 226 days (seven and a half months), and the median was 188 days (six months). However, over 6% of breakdowns in this study lasted over 420 days (13 months, representative of 7 herd-level tests, each 60 days apart, before OTF status was restored). Six variables associated with increasing breakdown length in cattle herds in NI were identified in both models. These can be grouped into three main categories; (1) variables related to herd characteristics, namely herd size and herd type; (2) variables related to undetected residual infection (i.e., infection within-herd), and (3) variables relating to local factors (i.e., infected wildlife, infected contiguous herds and a contaminated environment).

The MLVA genotype richness variable exhibited the strongest association with breakdown duration, both regarding effect size and in contribution to model fit. Previous work found that in a small number of herds, likely to be beef fattening enterprises, MLVA genotype accumulation was associated with the inwards purchase of cattle from over a wide geographical extent (Milne et al., 2019b). Despite this, we did not find that the number of inwards movements prior to breakdown was a particularly important predictor of breakdown length, e.g., (Reilly & Courtenay, 2007). However, we did not consider inwards cattle movements during a bTB breakdown, as businesses can be required to limit purchasing of cattle whilst bTB restricted, or where testing delays occur, banned from purchasing (a consequence of the bTB control program). Nevertheless, some beef fattening herds may indeed continue to purchase cattle despite the presence of bTB, as such enterprises operate by selling animals straight to slaughter (as opposed to onwards to other herds) and are only minimally impacted by movement restrictions. It is therefore likely that both the elevated MLVA richness and prolonged breakdown periods observed in beef fattening herds are associated with cattle purchases during breakdowns. However, in other herd types, the accumulation of MLVA genotypes may result in the absence of inwards cattle movements if herds are also exposed to infection from contiguous farms, infectious wildlife, or a contaminated environment. Given the spatial structuring of the M. bovis population (Skuce et al., 2010), we contend that it is more likely that re-infection from local sources would present with same M. bovis strains that are already present in the herd and local geographic area. Increasing MLVA richness would therefore have to involve the introduction of MLVA types from over a larger geographical extent. Whilst there is some evidence that badgers can occasionally travel long distances at scales of 7-20 km (Byrne et al., 2014a), it may be less likely that long-distance badger movements are an important source of MLVA richness relative to cattle movements, which can traverse national scales (Brown et al., 2019). The increased resolution provided by pathogen whole-genome sequencing (WGS), especially when more fully integrated with epidemiological data and modelling, may help to better understand transmission dynamics and the relative role of hosts in a multi-host system (Trewby et al., 2016).

Previous work from NI (Doyle et al., 2016), GB and the ROI (Clegg et al., 2018; Karolemeas et al., 2010; Karolemeas et al., 2011; Olea-Popelka et al., 2008; Wolfe et al., 2010) found that increasing herd size was positively associated with breakdowns lasting longer than 365 days. This may be related to the inability to detect all bTB-positive animals using the non-gold–standard ante-mortem SICCT test (Nuñez-Garcia et al., 2017). In NI, the relative sensitivity of the SICCT test may be as low as ∼40% in chronically infected herds (Lahuerta-Marin et al., 2018). Undetected animals, where present, represent an ongoing reservoir of residual infection which can lead to recrudescence of infection. The risk associated with herd size, however, may also be confounded with production type. Here, we found that herds with a milk license (i.e., dairy herds) were larger than herds without a milk license. Dairy farms may be associated with particularly intensive production, potentially increasing within-herd transmission (i.e., amplification) of infection (Alvarez et al., 2012; Menzies & Neill, 2000). Furthermore, there is some evidence that the SICCT test performs poorly in dairy in NI settings compared to beef (Lahuerta-Marin et al., 2018) which could exacerbate the problem presented by of residual infection. In the final multivariable models presented here, however, the presence of a milk license was not found to be an important predictor of breakdown duration. We hypothesize that other variables which differ between production types (e.g., herd size) have captured some important differences between animal husbandry practices which may be related to breakdown duration.

The number of reactors in the disclosing test was also positively associated with breakdown duration. We speculate that the presence of a large number of reactors at the disclosing test may indicate severity of infection, possibly arising from an environment which facilitates rapid within-herd transmission e.g., intensive farming units, or shared housing (Alvarez et al., 2012). Unless all animals infected with M. bovis are identified and removed from the herd as soon as possible, the rapid dissemination of infection will continue, thereby prolonging the outbreak duration (i.e., residual infection leading to within-herd recrudescence). Alternatively, many reactors at the disclosing test may indicate that infection has been either present or introduced since the preceding SICCT test, thereby providing a time period during which dissemination of infection to susceptible hosts within the herd could occur. We found that herd bTB history, measured by the presence of at least one previous breakdown in the study, was also associated with breakdown duration in the count model. Taken together, we hypothesise that a high number of disclosing reactors and a history of bTB indicates the presence of local infection (e.g., a contaminated environment, contiguous herds or infected wildlife), which may lead to increasingly prolonged outbreaks.

The presence of a lesioned animal at slaughter (LRS) was indicative of longer breakdowns in our models, which is in line with previous findings (Doyle et al., 2016). We argue that the presence of a tuberculosis lesion is often evidence of undetected bTB infection within the herd (Olea-Popelka et al., 2008). Indeed, previous work from NI confirmed that 97% of lesions from LRS animals were confirmed as bTB with histopathology or culture (Byrne et al., 2017). The relationship between bTB breakdown length and the presence of associated herds and elevated patch prevalence (Clegg et al., 2018; Doyle et al., 2016) illustrate the risk of infection from the local sources. Here, infection may originate from a shared contaminated environment (e.g., housing or grazing), which could lead to prolonged breakdowns if associated herds also contained infected animals. It may also point to shared use of equipment, or the spreading of contaminated slurry across multiple farms (O’Hagan et al., 2016). The positive relationship between local geography and prolonged breakdowns identified here has been observed previously in GB and the ROI (Olea-Popelka et al., 2008; Reilly & Courtenay, 2007). We suggest that geographical location variables (DVO and patch) are also a proxy for highly localised factors which could potentially influence breakdown length via exposure to other infected hosts in the area. These include degree of farm fragmentation, conacre use (shared grazing practice), and opportunities for contact with neighboring cattle (O’Hagan et al., 2016; White et al., 2013).

Wildlife and breakdown duration

In the univariable context, we identified a general positive relationship between breakdown duration and main sett density. Unsurprisingly, in the multivariate context, main sett density was confounded with other spatial and local variables (i.e., DVO and patch), making inferences on the contribution of badger density to infection prolongation less clear. It is not yet possible to conclusively distinguish between local sources of infection (which may include wildlife, contiguous herds and environmental contamination), but given our data, we cannot reject the hypothesis that badgers may be involved in the maintenance of local patch bTB prevalence via spillback infection to cattle. Whether infected badger presence has a greater risk of sporadic introduction of infection into herds (singular badger-cattle spillover), than longer-term maintenance within herds (explosive introduction of infection elevated with cattle-cattle transmission), remains to be determined. Whilst this study was unable to conclusively clarify the relationship between badger density and breakdown duration, our data nevertheless reveal important features that warrant further investigation in future studies.

Thus, despite DVO capturing the risk associated with main sett density and the general positive association between breakdown length and main sett density, there was some evidence of within-DVO effects. Within five DVO areas (Ballymena, Coleraine, Dungannon, Larne and Derry/Londonderry), increasing sett density was generally associated with longer breakdowns. In the other five DVO areas (Armagh, Enniskillen, Newry, Newtownards and Omagh), increasing sett density was generally associated with shorter breakdowns. The five DVOs with a positive association between sett density and breakdown duration were areas of generally lower badger sett densities (Reid et al., 2012). The DVOs with a negative association between sett density and breakdown duration were generally associated with higher badger sett densities. Whilst the interpretation of this is not straightforward, differences in farming practice (e.g., farm fragmentation) or differences in badger ecology (e.g., population context dependent badger dispersal; (Byrne et al., 2019) across the region could partially explain this observation. However, this does not preclude that the relationship between herd bTB risk from badgers may not be simply be dependent on wildlife density; the sett density data provides no insight regarding disease prevalence within the badger population. A spatially explicit model of disease prevalence in badgers may resolve this in future. Indeed, future research could investigate variation in wildlife TB transmission risk (LaHue et al., 2016) as a function of infection prevalence as well as density, and investigate how that could help to partially explain patterns within cattle data.

Conversely, infection risk in cattle may not be linked to badger disease prevalence or population density, but may instead related to the relative frequency of interactions between infected badgers and susceptible cattle (Böhm, Hutchings & White, 2009). Alternatively, it may be that indirect transmission of bTB via, for example, cattle accessing badger latrines, is more critically associated with chronic bTB breakdowns as opposed to wildlife population density per se (Campbell et al., 2019; Drewe et al., 2013). Furthermore, despite sett density being a convenient metric, we must be careful when inferring the relationship between sett density and population density, as the magnitude of the association can change depending on the local dynamics. For example, population density can increase without necessarily increasing the number of setts, via an increase in the mean group size (Judge et al., 2014). Alternatively, where badger population densities are depressed (e.g., though hunting, culling, or illegal disturbance), sett density metrics can overestimate true local density. In Ireland, sett density was found to be good predictor of increased herd breakdown risk early in a six-year study, but progressively became a weaker predictor as a program of targeted badger culling reduced population density (Byrne et al., 2014b). Therefore, investigating intricate relationships between wildlife and domestic hosts may well require even more detailed information around population abundance at large scales, in the Northern Ireland context this could include mark-recapture and/or the use of remote camera trapping technologies (Campbell et al., 2019).

Conclusions

The most important predictor of breakdown duration in our models was elevated MLVA genotype richness, which is often a feature of beef fattening herds and linked to the practice of purchasing cattle from over a wide geographic extent. We conclude that in at least some specific herds, prolonged restriction periods may primarily be a product of inwards cattle movements during a breakdown. For all other herd types, our results support the hypothesis that breakdown duration is principally a function of the inability to eradicate residual infection already present within the herd, and/or repeated infection from the local environment. In many instances, failure to clear residual infection may be related to the poor performance of the ante-mortem diagnostic SICCT test, which permits the retention of infected animals. Our data suggest that infected wildlife (captured by sett density), contiguous herds (captured by patch prevalence and associated herds) and a contaminated environment (also captured by patch prevalence) all likely contribute to varying extents to protracted breakdowns. However, given that it is not yet possible to positively distinguish between these various infection routes, determining the relative contribution each potential source was beyond the scope of this study. We posit that badgers may be involved in prolonging bTB breakdowns via spillback infection into the cattle population, supplemented with cattle-to-cattle transmission (amplification) once infection is introduced to the herd. However, the general positive association between badger sett density and breakdown duration may not simply be a function of badger population density, and could also be product of density-dependent badger behavior which may possibly influence contact rates between badgers and cattle.

Supplemental Information

Figure S1 Boxplots showing (A) the breakdown length (Y-axis) per-DVO (x-axis), and (B) the main sett density (Y-axis) per-DVO (X-axis). Data on the Y-axis have been log-transformed to improve visual inspection

Click here for additional data file.

Table S1 Results of the negative binomial count model of breakdown duration (untransformed model coefficients)

Click here for additional data file.

Table S2 Results of the negative binomial count model of breakdown duration, with log MLVA richness included with a quadratic term (untransformed model coefficients)

Click here for additional data file.

Table S3 Results of the ordinal model of breakdown duration (untransformed model coefficients)

Click here for additional data file.

Table S4 Results of (a) a negative binomial count model of breakdown duration and (b) a Gaussian GLM of breakdown duration (for illustrative purposes), with DVO and log main sett density as non-interacting predictors (untransformed model coefficients)

Click here for additional data file.

Table S5 Results of (a) a negative binomial count model of breakdown duration and (b) a Gaussian GLM (for illustrative purposes only) of breakdown duration, with DVO and log main sett density as interacting predictors

The interaction between log main sett and DVO was significant (negative binomial: χ2 = 24.24, df = 9, p = 0.004); untransformed model coefficients.

Click here for additional data file.

Table S6 The impact of main sett density on breakdown duration on a per-DVO basis, showing that in some DVOs, increases in main sett density is associated with increasing breakdown duration (Difference = positive)

In other DVOs, increases in main sett density is associated with decreasing breakdown duration (difference = negative). These numbers are extracted from Supplementary Table 4 (univariable model) and Supplementary Table 5 (full model). The data from the Gaussian GLM (as opposed to the negative binomial GLM) are presented for illustrative purposes and for ease of interpretation (untransformed model coefficients).

Click here for additional data file.

Table S7 Results of the negative binomial count model of breakdown duration, with DVO omitted from the random effects and log main sett included as a fixed effect, instead of log patch prevalence (untransformed model coefficients)

Click here for additional data file.

Table S8 Results of the negative binomial count model of breakdown duration, with DVO omitted from the random effects and log patch prevalence included in the fixed effects. Log main sett is included as a fixed effect (untransformed model coefficients)

Click here for additional data file.

Supplemental Information 1 Raw data

Click here for additional data file.

We would like to thank the staff in the Agri–Food and Biosciences Institute Veterinary Sciences Division (VSD) who contributed to the bTB strain–typing work.

Additional Information and Declarations

Competing Interests

Author Contributions

Animal Ethics

Data Availability

Andrew W. Byrne is an Academic Editor for PeerJ.

Georgina Milne conceived and designed the experiments, analyzed the data, prepared figures and/or tables, authored or reviewed drafts of the paper, and approved the final draft.

Adrian Allen, Carl McCormick, Eleanor Presho and Robin Skuce authored or reviewed drafts of the paper, generated molecular data, and approved the final draft.

Jordon Graham authored or reviewed drafts of the paper, managed and administered databases, and approved the final draft.

Angela Lahuerta-Marin authored or reviewed drafts of the paper, and approved the final draft.

Neil Reid authored or reviewed drafts of the paper, supplied wildlife data, and approved the final draft.

Andrew W. Byrne conceived and designed the experiments, authored or reviewed drafts of the paper, project management (PI), and approved the final draft.

The following information was supplied relating to ethical approvals (i.e., approving body and any reference numbers):

Data were generated as part of routine state lead animal health policy; no data were generated specifically for this project.

The following information was supplied regarding data availability:

The raw data is available in the Supplementary File and at Figshare: Milne, Georgina (2019): breakdown_length_anonymised_data. figshare. Dataset. https://doi.org/10.6084/m9.figshare.9777641.v2.

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
