# Peer review of "Bovine tuberculosis breakdown duration in cattle herds: an investigation of herd, host, pathogen and wildlife risk factors"

_PeerJ, doi:10.7717/peerj.8319_

## Round 0.1 · original submission · Minor Revisions

We would be grateful if you could address these minor comments before acceptance.

Reviewer 1 ·

Basic reporting

No comment.

Experimental design

No comment.

Validity of the findings

No comment.

Additional comments

This is very well written and careful analysis of risk factors associated with the duration of bovine TB restrictions applied to farms in Northern Ireland. Going beyond previous work, the authors explore the association between breakdown duration with badger population estimates and the within-herd diversity of M. bovis isolates.

The risk associated with genotype diversity is striking even given the potential confounding with within herd prevalence (which is carefully quantified and discussed).

The contribution of variables associated with badger population density is less clear and the authors have carefully explored the potential reasons for this and limitations of the data available for the study. However, the most obvious limitation of this data - namely that prevalence within the badger populations could also vary spatially independently of sett density should be acknowledged.

Minor comments:

Line 34, "breakdown" should be defined on first use. Not clear what the distinction being made here is as a breakdown is the terminology used to describe a herd (currently) affected by bTB.

Line 92, "...bTB breakdowns is not not well understood..." ?

Line 115: "(OTF) status." ?

Lin 192: "considered", extra space

Reviewer 2 ·

Basic reporting

The article is extremely well written, in a clear and unambiguous language.
The employed bibliography is adequate and updated.
Figures and tables are relevant for understanding the results. They are not superfluous. Raw data are available.
The work hypothesis are sustained by the undertaken analyses.

Experimental design

The presented work complies with the scope of the journal.
The research question is well defined, relevant and meaningful to the field. The article by Georgina Milne and colleagues is a particularly well written and clearly presented retrospective study using a very rich collection of meticulously selected and organised data from Northern Ireland breakdowns from January 2003 to December 2015.
It is stated how the presented research can help to identiy a knowledge gap.
The aim of the study was to better understand the factors associated with breakdown duration by objectivising well acknowledged bTB factors, using negative binomial and ordinal regression approaches. Different explanatory variables related to herd characteristics, to their residual infection or to local factors were considered. Six variables associated with increasing breakdown length were identified in both models: the herd size, the number of reactors on a first SICCT test, the presence of animals with bTB visible lesions at the abattoir, the local herd-level bTB prevalence, the presence of herds linked via management factors (associated herds) and, very strongly, the genotype (MLVA) richness during the breakdown.
The investigation is rigorous.
The methods are described with sufficient detail.

Validity of the findings

The findings are valid. The conclusions are sound and well supported by the results.
Very interestingly, the contribution of badger’s density (measured using sett metrics) to prolonged breakdowns was not clearly demonstrated in the study.

Additional comments

This a very insightful article that, without doubt, will help to improve bTB management guidelines in Northern Ireland.
Please, add an "is" before "not well understood" at line 92, and a "of" before "the longest breakdowns" at line 303.

---

## Round 0.2 · accepted · Accept

Dear authors, the reviewers' have recommended publishing of your article in the present form. Thank you for your interest in Peerj.
Kind regards,

Reviewer 2 ·

Basic reporting

Nothing to add from my previous review

Experimental design

Nothing to add from my previous review

Validity of the findings

Nothing to add from my previous review

Additional comments

Nothing to add from my previous review